# HNF1A Mutations and Beta Cell Dysfunction in Diabetes

**DOI:** 10.3390/ijms23063222

**Published:** 2022-03-16

**Authors:** Yasutaka Miyachi, Takashi Miyazawa, Yoshihiro Ogawa

**Affiliations:** Department of Medicine and Bioregulatory Science, Graduate School of Medical Sciences, Kyushu University, Fukuoka 812-8582, Japan; miyazawa.takashi.975@m.kyushu-u.ac.jp (T.M.); yogawa@med.kyushu-u.ac.jp (Y.O.)

**Keywords:** diabetes, MODY, MODY3, HNF1A, beta cell dysfunction, dedifferentiation, GWAS

## Abstract

Understanding the genetic factors of diabetes is essential for addressing the global increase in type 2 diabetes. HNF1A mutations cause a monogenic form of diabetes called maturity-onset diabetes of the young (MODY), and HNF1A single-nucleotide polymorphisms are associated with the development of type 2 diabetes. Numerous studies have been conducted, mainly using genetically modified mice, to explore the molecular basis for the development of diabetes caused by HNF1A mutations, and to reveal the roles of HNF1A in multiple organs, including insulin secretion from pancreatic beta cells, lipid metabolism and protein synthesis in the liver, and urinary glucose reabsorption in the kidneys. Recent studies using human stem cells that mimic MODY have provided new insights into beta cell dysfunction. In this article, we discuss the involvement of HNF1A in beta cell dysfunction by reviewing previous studies using genetically modified mice and recent findings in human stem cell-derived beta cells.

## 1. Introduction

The number of patients with type 2 diabetes (T2DM) has increased four-fold worldwide over the past 30 years, and is expected to increase by another 1.5 times in the next 20 years [1]. In dealing with the T2DM epidemic, it is crucial to consider the etiology and pathology of the disease.

T2DM is a multifactorial disease caused by both genetic and environmental factors, including diet, physical activity, and age. It is well recognized that obesity and aging contribute to diabetes development, however, there are two populations with the same weight and age: one with diabetes and the other with normal glucose levels. This difference in glucose tolerance is derived from genetic factors.

Genetic factors in diabetes have been investigated globally in multiple regions and races for (1) gene mutations in both nuclear and mitochondrial DNA, and (2) single nucleotide polymorphisms (SNPs) associated with diabetes. The most common form of monogenic diabetes is MODY, which accounts for 1–2.5% of all diabetes cases [2,3]. MODY is characterized by autosomal dominant inheritance, early onset of diabetes under 25 years of age, and impaired insulin secretion due to mutations in transcription factors involved in beta cell differentiation and function. To date, 14 MODY subtypes have been reported, including MODY1 (HNF4A), MODY2 (GK), MODY3 (HNF1A), MODY4 (PDX1), MODY5 (HNF1B), and MODY6 (NeuroD1) [4]. Among these, HNF1A gene mutations are the most common cause of MODY. In addition, SNPs in and near HNF1A are associated with an increased risk of T2DM.

Although an association between T2DM and HNF1A SNPs has been reported, it is unclear whether this relationship is causal, similar to that observed between MODY3 and HNF1A mutations. To clarify the pathophysiological role of HNF1A in diabetes, multiple studies have been conducted using Hnf1a-null mice; however, due to the inconsistency of phenotypes between Hnf1a-null mice and people with diabetes, the molecular basis has not been fully elucidated.

Undoubtedly, genetically modified animals will continue to be a powerful tool for elucidating human pathology. Nevertheless, recent human stem cell-derived beta cell research has provided new insights into the pathogenesis of diabetes caused by genetic variants in humans [5]. Here, we review the current understanding of HNF1A in multiple organs, including the pancreas, and discuss how HNF1A mutations contribute to the pathogenesis of diabetes, primarily focusing on beta cell dysfunction.

## 2. Extrapancreatic Organs and HNF1A

HNF1A is a transcription factor that belongs to the HNF1 homeobox family. The endogenous ligands of HNF1A are unknown, whereas aspirin and resveratrol have been identified as candidates for exogenous ligands by in silico analysis [6]. Aspirin and resveratrol are not specific ligands for HNF1A because they affect various molecules, including cyclooxygenases and sirtuins, respectively; however, these substances can potentially modulate the activity of HNF1A. Furthermore, protein–protein interactions have been identified between HNF1A and sirtuins [7,8]. For example, sirtuin 1 physically interacts with HNF1A in the liver to regulate HNF1A target genes [8].

HNF1A is mainly expressed in the liver, gut, pancreas, and kidneys. The physiological roles of HNF1A in each organ are summarized in Figure 1. During mouse embryo development, HNF1A mRNA was detected in the yolk sac at embryonic day (E) 8.5, and in the liver, intestine, and kidneys at E14.5 [9]. HNF1A has three isoforms: A (encoded by exons 1–10), B (exons 1–7), and C (exons 1–6). Isoform A is predominant in the liver, kidneys, and fetal pancreas, whereas isoforms B and C are abundant in the adult pancreas and islets [10].

### 2.1. Liver and HNF1A

HNF families, including HNF1A, are involved in liver development, function, and tumor growth [11]. For example, hepatocellular adenomas (HCAs), benign liver tumors, are classified into eight subgroups based on genotypes; one subgroup is defined by inactivating HNF1A mutations with a histological feature of tumor steatosis [12]. The majority of HCAs in this subgroup exhibit biallelic somatic inactivation of HNF1A. Certain HCAs found in MODY3 families contain a somatic mutation in one allele within the tumor and a heterozygous germline mutation in the other allele [13,14,15], showing the tumor-suppressive role of HNF1A in the liver.

Lipid metabolism plays a critical role in liver function. Homozygous Hnf1a-deficient mice exhibited growth retardation and hepatomegaly [16]. Hnf1a-null mice also had hyperbileacidemia and hypercholesterolemia, along with altered gene expression related to the synthesis and uptake of bile acids and de novo biosynthesis of cholesterol [17]. In another study, Hnf1a-null mice had elevated blood triglyceride levels and fatty liver accompanied by increased fatty acid synthesis [18]. Furthermore, lipidomic analysis of HCAs caused by HNF1A inactivation showed that de novo lipogenesis is enhanced in tumors, revealing the underlying mechanism of lipid accumulation in tumors [19]. However, the lipid profile of MODY3 and non-diabetic participants was similar, characterized by higher high-density lipoprotein cholesterol levels and lower triglyceride levels compared to T2DM participants [20,21]. Thus, heterozygous HNF1A mutations are unlikely to be sufficient to cause abnormal lipid metabolism.

Protein synthesis plays another important role in liver function. Albumin and C-reactive protein (CRP) are typical proteins produced by the liver. High CRP levels have been associated with various diseases, including cardiovascular disease [22], T2DM [23], obesity [24], psychological distress and depression [25], and cancer [26]. Plasma CRP levels are also associated with SNPs in and near the HNF1A gene [27]. Moreover, high-sensitivity CRP levels are lower in MODY3 groups than in a non-diabetic group and other diabetic groups, including type 1 and type 2 diabetes, MODY1, MODY2, and MODY5 [28,29]. There are two HNF1A-binding sites within the human CRP promoter and HNF1A is capable of activating CRP transcription [30]. Therefore, it is conceivable that the transcriptional activity of HNF1A is reduced in the liver of patients with MODY3.

### 2.2. Kidney and HNF1A

The expression of HNF1A and HNF1B was detectable in the kidneys during early development [9]. HNF1B mutations cause developmental kidney diseases, including renal cysts, single kidneys, renal hypoplasia, and electrolyte abnormalities [31], whereas the majority of HNF1A mutations are not associated with abnormal kidney morphology [32,33]. It has been observed that the reabsorption of urinary glucose is lower in patients with MODY3 than in those with type 1 and type 2 diabetes [34,35]. Intriguingly, the binding of phlorizin, a competitive inhibitor of sodium–glucose cotransporter (SGLT) 1 and 2, to the cell surface of proximal tubules was diminished in Hnf1a-null mice [16]. Furthermore, the SGLT2 expression was decreased in the kidneys of Hnf1a-null mice [35], and HNF1A directly promoted SGLT2 transcription in rodents and humans [35,36,37].

SGLT2 inhibitors have mainly been used for T2DM treatment. In one study, patients with MODY3 received a single dose of dapagliflozin, an SGLT2 inhibitor [38]. Surprisingly, changes in urinary glucose excretion following drug administration were more substantial in patients with MODY3 than in those with T2DM, contrary to the expectation that the drug effect would be attenuated in patients with MODY3, owing to the decreased SGLT2 expression. The precise reason for this result is unclear; however, it is possible that urinary glucose reabsorption via SGLT1, other than SGLT2, is enhanced in patients with T2DM [39,40,41].

Hnf1a-null mice also exhibited phenylketonuria and Fanconi syndrome [16]. However, these phenotypes were not reproduced in another study [42]. In addition, there have been few reports of renal dysfunction in patients with MODY3, whereas HNF4A mutations (MODY1) can cause Fanconi syndrome [43,44,45].

### 2.3. Gut and HNF1A

HNF1A and HNF1B are highly expressed in the crypts of the small intestine [46], and cooperatively regulate intestinal cell differentiation [46,47]. Mice with Hnf1a or Hnf1b deficiency in the gut had no severe phenotypes, however, mice with Hnf1a and Hnf1b double deficiencies in the gut died of dehydration soon after being born because of their inability to absorb water in the intestine [47].

Lactase-phlorizin hydrolase (LPH) is expressed in the small intestine, and plays an important role in the digestion of lactose. The physical interaction between the DNA-binding domain of HNF1A and the C-terminal zinc finger of GATA5 synergistically activates the promoter of human LPH [48].

The role of HNF1A in the gut in diabetes has not been well investigated; however, ghrelin, an appetite-stimulating hormone produced primarily in the stomach and intestine, is a potential target of HNF1A. Hnf1a-null mice possessed a higher number of ghrelin-positive cells in the intestine, accompanied by increased ghrelin gene expression, with elevated total and active ghrelin levels in the blood [49]. Likewise, fasting total ghrelin levels were elevated in patients with MODY3 compared to those with type 1 and type 2 diabetes [50]. In contrast, acylated ghrelin, or active ghrelin, was comparable in patients with MODY3 to healthy individuals and those with T2DM [51]. These inconsistent results may partly reflect differences in the measurement of ghrelin (i.e., total or acylated ghrelin).

## 3. Pancreas and HNF1A

### 3.1. Physiological and Pathophysiological Roles of HNF1A in Endocrine and Exocrine Cells of the Pancreas

HNF1A is expressed in endocrine cells, including alpha, beta, delta, and pancreatic polypeptide (PP) cells, and exocrine cells, including acinar and duct cells, in mouse pancreatic tissue [52]. HNF1A was detected in most pancreatic epithelial cells at E10.5, and hormone-positive cells and amylase-positive cells at E15.5, whereas the HFN1A expression was weak in duct cells [52]. During liver development, a transcriptional hierarchy exists, in which HNF4A positively regulates HNF1A expression [53,54], whereas HNF1A controls HNF4A by directly binding to the P2 promoter of HNF4A in differentiated pancreatic cells [55,56]. In addition, HNF1A regulates the PDX1 expression, which is essential for pancreatic development and the maintenance of beta cell function, in a cooperative manner with PDX1 itself in rodent beta cells [57]. The interaction between HNF1A and HNF4A is important in the differentiation of human induced pluripotent stem (iPS) cells into beta cells, supporting the relevance of the HNF1A–HNF4A axis during pancreatic development [58].

Pancreatic endocrine cells consist of alpha, beta, delta, PP, and other cells (e.g., ghrelin-producing cells). HNF1A was involved in glucagon secretion during hypoglycemia by directly regulating SGLT1 expression in mouse alpha cells [59]. Hnf1a-null mice showed impaired insulin secretion and subsequent hyperglycemia, suggesting that HNF1A was involved in insulin secretion in mouse beta cells [42,60]. However, heterozygous Hnf1a-deficient mice have normal blood glucose levels [60]. In addition, Hnf1a-null mice on a C3H or CBA background were non-diabetic [61], demonstrating phenotypic differences between patients with MODY3 and Hnf1a-null mice. The role of HNF1A in delta, PP, and other cells remains unknown.

Hnf1a-null mice showed a high acinar cell proliferation rate, and both endocrine and exocrine granules were observed in acinar cells, suggesting that HNF1A is required for normal exocrine differentiation [62]. Although it is undetermined whether HNF1A mutations lead to disorders in human exocrine cells, genome-wide association studies (GWAS) have revealed an association between HNF1A SNPs and pancreatic cancer [63,64]. Pancreatic cancer generally develops from the exocrine gland, and pancreatic ductal adenocarcinoma (PDAC) is the most common pancreatic cancer. A recent study has shown that HNF1A is highly expressed in pancreatic cancer stem cells (PCSCs), suggesting that HFN1A is a potential central regulator of PCSC function [65]. In that study, the knockdown of HNF1A in multiple PDAC cell lines resulted in tumor growth inhibition and apoptosis, accompanied by a reduction in stem cell markers. In contrast, HNF1A overexpression in PDAC cell lines promoted tumor growth, accompanied by the elevation of stem cell markers. These data indicate that the activation of HNF1A in pancreatic cancer promotes tumor growth, while HNF1A has also been shown to have a tumor-suppressive role in pancreatic cancer [66,67,68]. For example, HNF1A exerts its tumor-suppressive effect by forming a complex with lysine-specific demethylase 6A, encoded by the KDM6A gene, to inhibit the oncogenic pathway in a mouse acinar cell line [68]. These discrepancies may be due to the use of different cell lines in each experiment.

### 3.2. Type 2 Diabetes and the HNF1A Gene

HNF1A consists of 631 amino acids, including amino-terminal dimerization (residues 1–32), DNA-binding (203–276), and transactivation domains (281–631) [69]. The most frequent HNF1A mutation in MODY3 is P291fsinsC (p.G292fs), which has a dominant negative effect on wild-type HNF1A by lacking most of the transactivation domain [70,71,72]. Missense mutations are abundant in the amino-terminal dimerization and DNA-binding domains, and deletion mutations are frequently found in the transactivation domain [73,74], potentially leading to clinical heterogeneity in patients with MODY3 [75,76].

SNPs are the most common DNA sequence variants in the genome, with an estimated 10 million SNPs in the human genome [77]. Statistical analysis of the association between diseases and SNPs allows us to infer disease risk, the molecular basis of the disease, and drug targets for patients with certain SNPs. GWAS have identified over 100 common variants that cause T2DM in humans (minor allele frequency (MAF) > 1%) [78]. According to the simulated models of T2DM, these common T2DM variants can potentially account for approximately 75% of the heritability of the disease [79]. The individual common variants are weakly associated with T2DM (odds ratio < 1.3), however, people with more disease-associated variants are expected to have an increased risk of developing the disease [80]. In contrast, rare coding variants (MAF < 0.5%) have a negligible effect on the heritability of T2DM [81]. Considering the small contribution of rare variants, the common variants appear to play a significant role in the heritability of T2DM.

HNF1A SNPs are associated with the risk of T2DM, and this finding was confirmed across different ethnic groups [82,83,84]. In detail, rs1169288 (I27L) [85,86], rs1800574 (A98V) [85,86], rs140730081 [85,86], G319S [87], rs2464196 (S487N) [88], M490T [89], E508K [89], rs7957197 [82], and rs12427353 [90] were associated with T2DM. Among these, I27L, A98V, and S487N are common coding variants of HNF1A, and may have a small impact on the risk of T2DM [91,92].

An oral glucose tolerance test (OGTT) was performed on 17 participants with rare coding variants of HNF1A (MAF < 1%), and only one showed an aberrant increase in 2-h glucose levels after 75 g OGTT (defined as a difference of >90 mg/dl between 2-h glucose and fasting glucose levels) [93]. Although insulin concentrations were not measured in these rare variant carriers, it is possible that the reduction in insulin secretion was partly compensated by an increase in insulin sensitivity and a decrease in the threshold for urinary glucose excretion, as observed in patients with MODY3 [88,94]. Furthermore, the results suggest that the functional validation of HNF1A variants is necessary [91], and also demonstrate that the prediction of T2DM in the general population based on rare variants obtained from GWAS can overestimate the contribution of rare variants, leading to an increase in the incidence of false positives [93,95]. Rare variants of a disease are generally assumed to have a significant effect on the phenotype, however, potential ascertainment biases in Mendelian genetic studies occur, even for rare variants of HNF1A that have been shown to cause MODY3 owing to the selection of patients with overt diabetes or a family history of MODY [76,93,95].

Despite the frequency of MODY3 and the association of HNF1A SNPs with T2DM, the molecular basis of HNF1A mutations leading to diabetes remains elusive. In the following sections, we will discuss how HNF1A controls insulin secretion and regulates downstream genes in beta cells.

### 3.3. Roles of HNF1A in Beta Cells

The microarray analysis of pancreatic islets from Hnf1a-null mice showed that HNF1A regulated the genes involved in glucose and amino acid metabolism, including glycolysis, tricarboxylic acid cycle, and oxidative phosphorylation [96]. MODY3 is characterized by reduced insulin secretion before the onset of diabetes and displays a distinct phenotype from type 1 and type 2 diabetes [97,98]. An impaired insulin secretion response to high glucose or arginine was revealed in ex vivo islets from Hnf1a-null mice [60]. The reduction in arginine-induced insulin secretion was also observed in MIN6 cells, a mouse beta cell line, expressing a dominant negative mutant of HNF1A (P291fsinsC) [99]. Human iPS cell-derived beta cells with a MODY3 mutation (HNF1A^+/H126D^) showed impaired glucose-stimulated insulin secretion (GSIS) after six months of maturation by transplantation into mice [100]. In addition, impaired GSIS due to HNF1A dysfunction was confirmed in islets from a diabetic patient harboring HNF1A^+/T260M^ [101].

T2DM affects insulin secretion through a combination of genetic factors, including the accumulation of disease-associated common variants, and environmental factors, including metabolic overload and aging. HNF1A mutations (MODY3) reduce insulin secretion by altering glucose metabolism in beta cells [96,102,103]. HNF1A mutations result in decreased glucose transporter 2 (GLUT2) and glucose uptake [100,102]. HNF1A regulates liver-type pyruvate kinase, a rate-limiting enzyme in glycolysis, and GLUT2 in rodent beta cells [104,105,106,107]. Furthermore, the most common HNF1A mutation in MODY3 (P291fsinsC) reduces mitochondrial ATP production in mouse beta cells [107]. Therefore, HNF1A is likely to be involved in ATP production, predominantly in the mitochondria, and subsequent insulin secretion, by maintaining glucose flux (Figure 2).

It is controversial whether HNF1A directly regulates mitochondrial function-related genes; these genes might be regulated through a network of transcription factors, such as PDX1 and HNF4A [104]. Loss of HNF1A in human embryonic stem (ES) cell-derived beta cells impairs mitochondrial respiration accompanied by a decrease in LINKA, a human-specific long non-coding RNA, suggesting the involvement of HNF1A in mitochondrial function [108]. Further studies are required to elucidate the role of HNF1A in regulating mitochondrial metabolism in beta cells.

In addition to the central metabolic pathway, HNF1A directly regulates TMEM27 [109,110] and hepatocyte growth factor activator [111], which promote proliferation in mouse beta cells. Collectrin (TMEM27) also controls insulin exocytosis through the formation of the SNARE complex [110]. In addition, HNF1A regulates genes whose functions in beta cells are poorly understood, including AKR1C19 [96,112].

### 3.4. Epigenetics and HNF1A in Beta Cells

GWAS have provided us with a better understanding of the heritability of T2DM over the past 15 years; however, improving the accuracy of personalized risk prediction for T2DM remains a challenge [113,114]. The difficulty in accurately predicting the onset of T2DM is likely due to a complex combination of genetic and environmental factors, including individual lifestyle [115]. Moreover, much remains to be explored in terms of genetic factors. For example, epigenetic changes, including histone modification, DNA methylation, and non-coding RNA, can alter transcription factor binding to the genome and gene expression, independently of the DNA sequence. Although there is accumulating evidence that epigenetic changes are associated with the development of T2DM [116,117,118], the underlying molecular mechanisms are not fully understood.

Histone modification plays an important role in beta cell development and function [119]. The acetylation of histones loosens the chromatin structure and facilitates the binding of transcription factors to DNA, and is thus considered an active epigenetic marker. CREB-binding protein (CBP) and P300 are the major histone acetyltransferases; they have similar structures and are involved in the acetylation of H3K27. HNF1A is required for the maintenance of histone acetylation in the promoter regions of its target genes, including GLUT2, in mouse islets [105]. It was also found that CBP and P300/CBP associated factor interacted with the N- and C-terminal domains of HNF1A, respectively [120]. In addition, HNF1A recruits P300 to the promoter of human GLUT2 [121]. Interestingly, the genes that decreased in CBP- or P300-null islets overlapped with those in Hnf1a-null islets [122]. Taken together, these results indicate that HNF1A and CBP/P300 cooperate to activate gene expression, possibly through histone modification, including the acetylation of H3K27.

### 3.5. Beta Cell Mass and HNF1A

Beta cell dysfunction is a pathological hallmark of T2DM, and the preservation or restoration of beta cell mass is important for T2DM treatment. The decrease in beta cell mass in T2DM was considered to be caused by beta cell death [123]. However, within five years of diagnosing T2DM, the difference in beta cell mass between healthy individuals and patients with T2DM was small, with a large overlap, suggesting that the decrease in beta cell mass was a result of disease progression rather than a contributor to the development of the disease [124,125].

Although the data on beta cell mass in patients with MODY3 are inconclusive, beta cells from a patient with HNF1A^+/T260M^ showed impaired GSIS but no obvious decrease in the cell mass compared to the average beta cell mass of healthy individuals [101]. The beta cell mass in Hnf1a-null mice was smaller than that in heterozygous Hnf1a-deficient mice, and this difference disappeared after adjusting for body weight [60]. Alternatively, the common HNF1a mutation (P291fsinsC) decreased mouse beta cell proliferation [126]. Likewise, a dominant negative mutant of HNF1A increased sensitivity to endoplasmic reticulum (ER) stress in mouse islets and promoted drug-induced apoptosis in mouse beta cells [127,128]. In summary, HNF1A mutations may not have a significant impact on beta cell mass per se, however, it is possible that certain triggers, including metabolic stress, reduce beta cell proliferation and cause beta cell death in patients with HNF1A mutations.

Beta cell proliferation in adults is considerably low under normal conditions [129]; however, obesity expands beta cell mass in response to increased insulin demand [130]. Various explanations have been proposed for changes in beta cell mass, including beta cell replication [131,132,133], neogenesis [134,135], and transdifferentiation [134,136], which have been linked to the therapeutic promise of beta cell regeneration [137].

Cellular plasticity has been shown in the islets of patients with diabetes [125,138]. In Korea, a decrease in beta cell mass and an increase in the relative volume of alpha cells in islets were observed in non-obese patients with T2DM [139]. The mechanism underlying the increase in alpha cells in the islets of T2DM is unknown, however, the concept of beta cell dedifferentiation has recently been proposed [140]. Beta cell dedifferentiation is an adaptive process in which beta cells return to a progenitor-like stage after prolonged metabolic stress, and a portion of the immature beta cells convert to other endocrine cells, such as alpha and delta cells [141]. The concept of beta cell dedifferentiation explains the mechanisms of beta cell decline and alpha cell increase in T2DM [139,142,143,144] and supports the observation that beta cell loss progresses after the onset of T2DM [124]. In contrast, dedifferentiation plays a minor role in beta cell decline in obese T2DM patients with an average BMI > 40 [145]. Moreover, it remains unclear whether beta cell differentiation and ER stress-induced apoptosis can coexist in the same cells, or whether they are mutually exclusive.

The molecular basis of dedifferentiation remains to be fully elucidated [146], however, it has been shown that FoxO1 migrates to the nucleus in beta cells prior to dedifferentiation, maintaining the MODY network, including HNF1A and HNF4A [147]. The molecular basis of MODY-related genes causing diabetes and their involvement in dedifferentiation have rarely been investigated because of the difficulties in obtaining pancreatic tissue or beta cells from patients with MODY. Recent studies have attempted to overcome these obstacles by generating iPS cell-derived beta cells from MODY [100,148,149,150]. Studies using stem cell-derived beta cells have indicated a possible involvement of HNF1A in beta cell dedifferentiation [108,151]. HNF1A deficiency in human ES cell-derived beta cells results in increased alpha cell markers, including glucagon, while decreasing the expression of PAX4, which is involved in beta cell development [108]. Decreased PAX4 levels were also observed in iPS cell-derived beta cells lacking HNF1A [151]. Despite this evidence, the most compelling approach to the involvement of HNF1A in dedifferentiation is to determine whether beta cell dedifferentiation occurs in the pancreatic tissue of patients with MODY3. Interestingly, the alpha cell mass was higher in a diabetic patient harboring HNF1A^+/T260M^ than in healthy individuals [101]. It is currently unknown whether common variants associated with T2DM are involved in dedifferentiation. The roles of HNF1A variants in MODY3 and T2DM are summarized in Table 1.

## 4. Pharmacologic Treatment for Diabetic Patients with HNF1A Mutations

The glucose-lowering effect of metformin was comparable in patients with MODY3 and those with T2DM, while the sensitivity to sulfonylureas (SU) was higher in patients with MODY3 than in those with T2DM [152,153]. The high sensitivity of patients with MODY3 to SU may be due to their increased insulin sensitivity [94]; therefore, patients with MODY3 are likely to be more susceptible to hypoglycemia with SU [154].

Monotherapy with glucagon-like peptide-1 receptor agonists (GLP-1 RA) [153,155] or dipeptidyl peptidase-4 (DPP4) inhibitors [156,157] may prevent hypoglycemia and achieve good glycemic control in patients with MODY3. Moreover, combination therapy with GLP-1 RA and SU may increase insulin secretion in patients with MODY3 [158], and combination therapy with DPP4 inhibitors and SU may improve glycemic control without significantly increasing hypoglycemia in patients with MODY3 [154].

Given the response to SU treatment for patients with MODY3, it is possible that HNF1A mutations do not substantially impair the insulin secretory pathway after membrane depolarization in beta cells. However, HNF1A mutations have been shown to alter the formation of mature insulin secretory granules [110,151,159], and HNF1A may also regulate insulin exocytosis. In addition, the impact of HNF1A mutations on other endocrine cells, including alpha cells, requires further investigation in humans.

## 5. Concluding Remarks

Recent findings indicate that HNF1A mutations are associated with an increase in alpha cells in human pancreatic islets and differentiation of stem cell-derived beta cells toward alpha cells, showing the relevance of studies using human islets and beta cells. The molecular basis of HNF1A abnormality in insulin secretion in human beta cells and the pathophysiological role of HNF1A in the liver, kidneys, and gut in diabetes requires further investigation. In this review, we discuss the similarities and differences in the pathophysiological role of HNF1A in patients with MODY and T2DM, however, few reports were found directly comparing the effects of HNF1A mutations in MODY3 and T2DM-related HNF1A common variants. Therefore, it is unclear whether the findings of HNF1A mutations in MODY3 can be extrapolated to the pathogenesis of T2DM. It is expected that in addition to conventional animal experiments, human stem cells and new technologies will reveal new molecular bases and therapeutic targets for diabetes (Table 2).

## Figures and Tables

**Figure 1 ijms-23-03222-f001:**
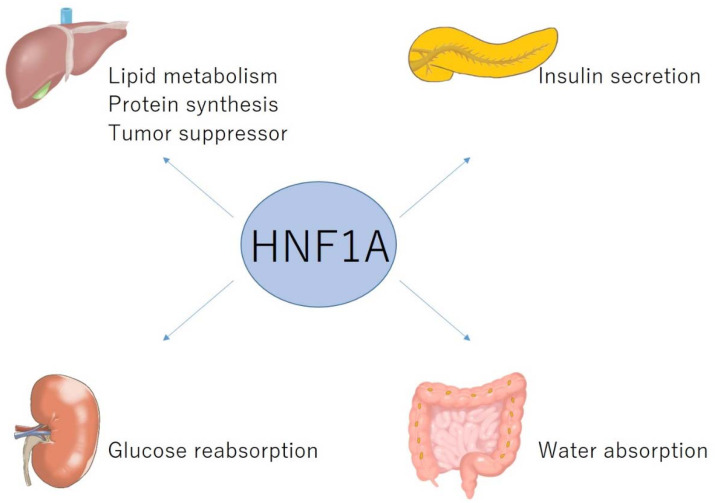
Physiological role of HNF1A in the liver, pancreas, kidneys, and intestine.

**Figure 2 ijms-23-03222-f002:**
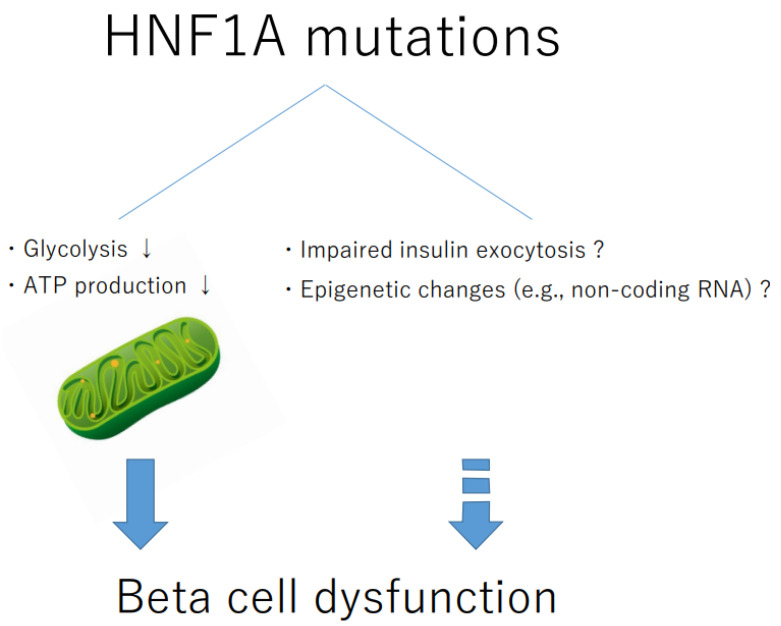
Schematic representation of beta cell dysfunction caused by HNF1A mutations.

**Table 1 ijms-23-03222-t001:** Comparison of the roles of HNF1A variants in MODY3 and T2DM. hs-CRP, high-sensitivity CRP; SU, sulfonylureas.

	HNF1A Variants	Types of Variants	Insulin Sensitivity	Beta Cell Fate	Sensitivity to SU
MODY3	Pathogenic role	Missense [73,74] and deletion [70,71,72,73,74] mutations	Low hs-CRP levels [28,29] Low renal threshold for glucose [34,35]	Reduced insulin secretion [97,98] Dedifferentiation [108,151]	High [152,153]
T2DM	Disease risk	SNPs [82,83,84,85,86,87,88,89,90]	Not reported	Certain SNPs (e.g., coding variants) may affect insulin secretion [91,92]	Not reported

**Table 2 ijms-23-03222-t002:** Future research directions.

	Aims	Objectives
Functional analysis of HNF1A variants	Molecular mechanism Therapeutic targets	Mainly use human stem cell-derived beta cells with HNF1A variants.
Insulin secretion		Assess insulin secretion, insulin granules, and mitochondrial metabolism in beta cells.
Epigenetics		Investigate changes in histone modification, DNA methylation, and non-coding RNA in beta cells.
Beta cell dedifferentiation		Confirm a decrease in beta cell mass and an increase in alpha cell mass in the pancreatic tissue harboring HNF1A variants.
Extrapancreatic organs		Measure hs-CRP levels and urinary glucose reabsorption in individuals with HNF1A SNPs.
Treatment for patients with HNF1A variants	Personalized medicine	Assess the sensitivity to SU using iPS cell-derived beta cells from individuals with HNF1A SNPs.

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
