# Peer review of "HNF1A Mutations and Beta Cell Dysfunction in Diabetes"

_ijms, 2022, doi:10.3390/ijms23063222_

Round 1

Reviewer 1 Report

This review is updated summary HNF1A genetics and pathophysiology. This manuscript can be accepted with minor revise.

  1. The authors add more information about HNF1A protein such as its domain and binding partner at different tissue.
  2. In line 62, HNA1A should be HNF1A.

Reviewer 2 Report

This is a review article regarding HNF1A mutations and beta cell dysfunction in diabetes. Overall, the manuscript was concisely and well written. The topic is of interest and informative. There are some comments.

  1. The authors discuss the similarities and differences in the role of HNF1A mutations between MODY3 and T2DM. It will be better if the authors summarize them in a table.

  1. It will be better if the authors summarize perspectives on future research direction in a table.
